# Bridging Domain Adaptation and Graph Neural Networks: A Tensor-Based Framework for Effective Label Propagation

Tao Wen[1]*, Elynn Chen[2], Yuzhou Chen[3], Qi Lei[2]

[1]Dartmouth College [2]New York University [3]University of California, Riverside

`tao.wen.gr@dartmouth.edu, elynn.chen@nyu.edu, yuzhou.chen@ucr.edu, ql518@nyu.edu`

Graph Neural Networks (GNNs) have recently become the predominant tools for studying graph data. Despite state-of-the-art performance on graph classification tasks, GNNs are overwhelmingly trained in a single domain under supervision, thus necessitating a prohibitively high demand for labels and resulting in poorly transferable representations. To address this challenge, we propose the Label-Propagation Tensor Graph Neural Network (LP-TGNN) framework to bridge the gap between graph data and traditional domain adaptation methods. It extracts graph topological information holistically with a tensor architecture and then reduces domain discrepancy through label propagation. It is readily compatible with general GNNs and domain adaptation techniques with minimal adjustment through pseudo-labeling. Experiments on various real-world benchmarks show that our LP-TGNN outperforms baselines by a notable margin. We also validate and analyze each component of the proposed framework in the ablation study.

## 1. Introduction

Graph data are ubiquitous in a wide variety of fields, with many objects such as chemical compounds, molecules, and social networks that are naturally represented by graphs [1]. The problem of graph classification [2, 3], which aims to predict the properties of whole graphs, is essential in such fields [4]. In recent years, Graph Neural Networks (GNNs) have served as the predominant tool to tackle this problem [2, 5–8]. The bulk of GNNs follow the message-passing framework [5, 9], which involves propagating and aggregating information of centroid nodes and their topological neighbors. They have consistently achieved superior performance on graph learning tasks, thanks to their adaptability, scalability, and capacity of graph representation learning.

Despite their advances, the overwhelming majority of GNNs are trained in a single domain under supervision [2, 10], thus demanding a sufficient number of labels that are often prohibitively expensive [11–13]. For example, labeling graphs of gene interactions is extremely expensive due to the need for expert knowledge, complex data analysis, data quality control, and adherence to ethical and privacy regulations. A consequent approach is domain adaptation, i.e. to train a graph classifier that works well on an unlabeled target domain with a labeled source domain.

Although domain adaptation methods have been extensively studied in computer vision [14–19], such methods for graph classification are challenging to develop due to two fundamental problems: 1) *How to effectively extract holistic graph representations?* Though GNNs have demonstrated superior performance in most cases, they are intrinsically unable to fully capture relevant connectivity information, such as higher-order interactions between atoms and ring-ring interactions within a molecule that are essential in drug discovery [20–22]. In contrast to node or link prediction problems where all samples are from a single graph [23], we are faced with a large number of graphs from distinct feature spaces and domains in graph classification. It is therefore crucial to learn high-quality representations that holistically encompass graph information and are well located in the embedding space for transfer learning, as illustrated in Figure 1. 2) *How to reduce domain discrepancy for*

---

*Work performed while at NYU.

Second Conference on Parsimony and Learning (CPAL 2025).

*graphs?* While domain adaptation methods are well-studied in computer vision, their applicability often relies on assumptions such as selection bias (suited for unstructured data) or environmental changes (attributing spurious correlations to factors like background information) [17, 24–26]. These assumptions, however, are not directly transferable to structured data such as graphs.

To address the challenges above, we propose to study the general framework that consists of 1) a graph encoder that extracts holistic graph representation and preserves graph similarities under its geometric distance, thus 2) readily and effectively integrates with certain domain adaptation methods. Under the general framework, we propose a model named **L**abel-**P**ropagation **T**ensor **G**raph **N**eural **N**etwork (LP-TGNN). Specifically, our choice of graph encoder TGNN partially stems from the recent TTG-NN [28] and consists of two branches that learn graph information both locally and globally. On the one hand, a graph convolutional branch follows the message-passing framework and stacks outputs of different graph convolutional network layers [5] to enhance its expressive power. On the other hand, a topological learning branch combines persistent images from multiple filtration functions of Persistent Homology (PH), thus extracting the graph topological information from multiple perspectives. This design is achievable thanks to a Tensor Transformation layer (TTL) that seamlessly integrates multiple attributes

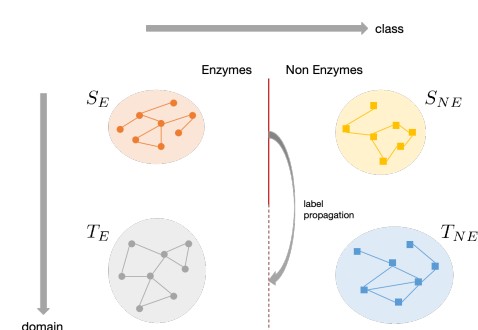

Figure 1: A visualization of domain adaptation on graph classification. The red line is the decision boundary of an Enzyme versus Non-Eyzyme classifier trained on $S_E$ and $S_{NE}$. The labels can transfer from the source domain to the target domain via methods such as label propagation [26]. For effective label propagation between domains, the intra-class distance is supposed to be smaller than the inter-class distance [27]. For instance, $T_E$ should be closer to $S_E$ than to $S_{NE}$.

while efficiently preserving the discriminative features by tensor low-rank decomposition. We investigate how different graph embeddings measure graph discrepancies in Euclidean distance with visualization in Figure 2. The scatterplots show that TTG-NN preserves in-group distance better than the expressive GIN [2], regardless of the domains. Such findings motivate us to utilize TTG-NN to convert the structured graph information to vector values that general domain adaptation techniques will apply.

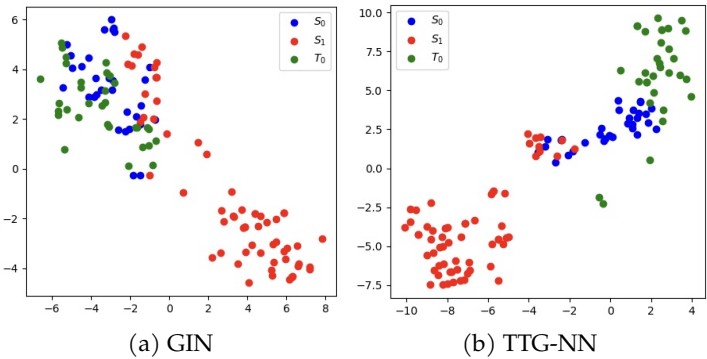

(a) GIN      (b) TTG-NN

Figure 2: An illustration of representations produced by the TTG-NN and the state-of-the-art GIN [2] on the MUTAG dataset [29]. While both graph encoders separate the three clusters effectively, TTG-NN well preserves the in-group distance with green points ($T_0$) and blue points ($S_0$) and pushes away data from a different group: red points ($S_1$). Embeddings plotted using t-SNE [30].

A substantial caveat for adversarial-based domain adaptation methods [24, 31] is that they can lose critical discriminative information on the target domain by forcing domain-invariant representations.

On the contrary, label propagation [26] reduces domain discrepancy through regularization. It encourages the predictions in a set of neighboring data samples to be stable via consistency regularization. With well-spaced representations, as shown in Figure 1 and 2, the label information can propagate from the source domain to the target domain. However, it requires well-defined sample distances for the label information to propagate within neighbor sets. While such distances are easy to define in computer vision tasks, they are ambiguous for structured data like graphs. Rather than directly computing such a distance, we instead define the neighboring set of a target graph to be the source graphs whose *ground-truth labels* are the same as the *pseudo-label* of the target graph, where the pseudo-label is given by a multi-layer-perceptron (MLP) classifier.

In conclusion, our method significantly differs from existing domain-adaptive GNNs as it works in the graph space rather than the node space, takes advantage of expressive GNNs and adapts regularization-based domain adaptation methods without stringent mathmatical formulations or forcing invariant representations across domains. Our main contributions are summarized as follows:

- We introduce a framework named LP-TGNN for domain adaptive graph classification that bridges the gap between computer vision domain adaptation methods like label propagation and structured data like graphs. We show that the graph learning ability of TGNN endows its potential to capture the relevance of graphs on the tensor embedding space.

- We design a label propagation scheme to reduce domain discrepancy for graphs. As a regularization method rather than a conventional distribution matching method, label propagation performs well with graph data and can be readily integrated with existing GNNs besides TGNN, resulting in a general framework.

- Experiments on various graph classification benchmarks demonstrate the effectiveness of our proposed framework and showcase the potential of extending domain adaptation methods such as label propagation to domain adaptive graph classification tasks.

## 2. Related Work

### 2.1. Graph Classification

Graph classification has enormous applications in various fields [32]. Traditional kernel methods use graph decomposition to capture the similarity in graph sub-structures with specialized kernels. For example, WeisfeilerLehman [33] proposes a family of kernels for large graphs with discrete node labels. Recently, Graph Neural Networks have emerged as a primary tool for graph classification [34]. For instance, Graph Convolutional Network [35] updates node representations iteratively by their neighboring nodes. GraphSAGE [6] performs inductive learning by aggregating information from local neighborhoods. GAT [36] uses the attention mechanism to capture dependencies between nodes. To handle large-scale graphs, Top-$K$ pooling [37–39] filters the nodes by their importance scores. Recently, PersLay [40] and RePHINE [41] utilize Persistent Homology to capture graph topological features. Their common limitations are: 1) They fail to exploit the rich semantic and topological graph information holistically from a multi-modal perspective, which is well-addressed in TTG-NN [28]; 2) They overwhelmingly focus on learning from a single domain and in a supervised manner, resulting in poorly transferable representations and prohibitive demand for graph labels.

### 2.2. Domain Adaptation

Domain adaptation aims to develop models that are transferable from a label-rich source domain to a label-scarce target domain [42, 43]. It has been profoundly investigated for image data with applications including image classification and semantic segmentation [14, 15, 19], and also explored for question-answering models [44, 45]. More complex settings such as multi-source [46] and multi-target [47] domain adaptation have also been studied. In short, the key to domain adaptation is domain alignment. Traditional statistical methods achieve this by explicitly reducing domain discrepancy via statistics such as maximum mean discrepancy [48, 49]. Recently, the conventional

approaches have been based on adversarial learning [31, 47, 50]. These methods typically employ a Gradient Reversal Layer (GRL) to force domain-invariant representations from the feature extractor [24] and utilize pseudo-labeling to allow for self-training [51]. However, while such methods for computer vision have been extensively studied, those for whole graphs are still in the infant stage.

## 2.3. Graph Domain Adaptation

Recently, a few methods have been proposed to address domain adaptation on graphs, mostly for node classification [23, 52–55]. However, for graph classification, graphs reside in diverse feature spaces instead of a unified space from a single graph as in node classification, leading to a much more challenging problem setting. To tackle this problem, DEAL [56] utilizes adversarial learning for domain alignment and distillation for pseudo-labeling. Furthermore, CoCo [57] advances from adversarial learning to contrastive learning, which consists of coupled branches for graph representation learning and contrasts between branches and domains. DAGRL [58] uses a similar architecture combined with adaptive adversarial perturbation to align source and target domains. Though the problem definition of domain adaptive graph classification has been established in these works, the problem remains under-explored and demands more effective approaches.

# 3. Preliminaries

**Problem formulation.** Suppose an attributed graph $\mathcal{G} = (\mathcal{V}, \mathcal{E}, \mathbf{X})$, in which $\mathcal{V}$ is the set of nodes, $\mathcal{E} \in \mathcal{V} \times \mathcal{V}$ is the set of edges, and $\mathbf{X} \in \mathbb{R}^{N \times F}$ is the feature matrix of nodes, where $N = |\mathcal{V}|$ is the number of nodes and $F$ is the dimension of node features. The adjacency matrix $\mathbf{A} \in \mathbb{R}^{N \times N}$ is a symmetric matrix with entries as $a_{ij} = 1$ if an edge exists between nodes $i$ and $j$ and 0 otherwise. Furthermore, $\mathbf{D}$ is the degree matrix of $\mathbf{A}$ with entries as $d_{ii} = \sum_j a_{ij}$.

In the setting of domain adaptive graph classification, we are given a source domain $\mathcal{D}^s = \{(\mathcal{G}_i^s, y_i^s)\}_{i=1}^{n_s}$ with $n_s$ labeled samples and a target domain $\mathcal{D}^t = \{\mathcal{G}_j^t\}_{j=1}^{n_t}$ with $n_t$ unlabeled samples. $\mathcal{D}^s$ and $\mathcal{D}^t$ share the same label space $\mathcal{Y} = \{1, 2, \cdots, C\}$ with covariate shifts, i.e. $\mathbb{P}_{\mathcal{D}^s}(\mathbf{X}) \neq \mathbb{P}_{\mathcal{D}^t}(\mathbf{X})$ and $\mathbb{P}_{\mathcal{D}^s}(y|\mathbf{X}) = \mathbb{P}_{\mathcal{D}^t}(y|\mathbf{X})$ [59]. The goal is to learn a graph classification model on $\mathcal{D}^s, \mathcal{D}^t$ that predicts the ground-truth labels in $\mathcal{D}^t$ accurately.

**Tensor low-rank structures.** The Tucker, CP, and Tensor Train (TT) low-rank are the conventional tensor low-rank structures. Specifically, the Tucker low-rank structure is defined by

$$\boldsymbol{\mathcal{X}} = \boldsymbol{\mathcal{C}} \times_1 \boldsymbol{U}_1 \times_2 \cdots \times_M \boldsymbol{U}_M + \boldsymbol{\mathcal{E}}, \tag{1}$$

where $\boldsymbol{\mathcal{E}} \in \mathbb{R}^{D_1 \times \cdots \times D_M}$ is the noise tensor and $\boldsymbol{\mathcal{C}} \in \mathbb{R}^{R_1 \times \cdots \times R_M}$ is the latent core tensor of the true low-rank features, and $\boldsymbol{U}_m, m \in [M]$ are the loading matrices. CP low-rank is a special case where the core tensor $\boldsymbol{\mathcal{C}}$ has the same dimensions over all modes. TT low-rank is a different kind of low-rank structure that can compress tensors as significantly as CP low-rank while maintaining a stable calculation as Tucker low-rank. With tensor low-rank structures, the Tensor Transformation Layer tackles model complexity and computation concerns that could arise from amalgamating diverse tensor features while preserving the discriminative features.

**Persistent Homology.** Persistent Homology (PH) is a method from algebraic topology that studies the shape of data across different scales. Given a scale parameter $\epsilon$, PH builds a sequence of simplicial complexes $\mathcal{G}_{\epsilon_1}, \mathcal{G}_{\epsilon_2}, \ldots, \mathcal{G}_{\epsilon_n}$, where each $\mathcal{G}_{\epsilon_i}$ represents the shape at a specific resolution. As $\epsilon$ increases, new features are "born," and some "die" when they merge into larger structures. These changes are recorded in a Persistence Diagram (PD), which is a set of points $(b_\rho, d_\rho)$ in $\mathbb{R}^2$. Each point represents a topological feature $\rho$, where $b_\rho$ is the birth time and $d_\rho$ is the death time. The difference $d_\rho - b_\rho$ is the lifespan of the feature, showing how long it persists across different scales. PD can be further converted to Persistent Image (PI) for computational benefits as detailed in appendix A.1.

# 4. Methodology

## 4.1. Overview

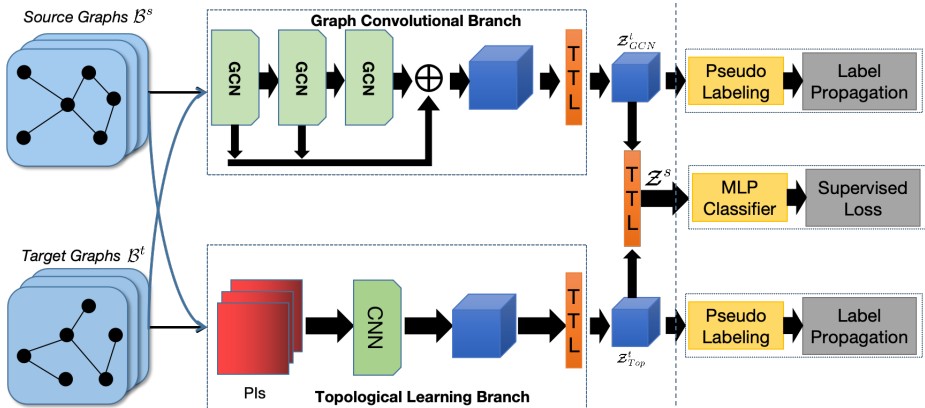

Figure 3: The architecture of the proposed LP-TGNN. The graph mini-batches from both domains are input into both branches. The representations of target samples from each branch are pseudo-labeled by an MLP classifier and each branch is separately regularized by label propagation. Also, the representations of source samples from both branches are concatenated and labeled by the MLP classifier. The model is optimized by the supervised loss and label propagation jointly.

Figure 3 illustrates the architecture of the proposed LP-TGNN. To address the two fundamental problems in Section 1, the model can be decomposed into two components: 1) The backbone encoder of TGNN and 2) The domain alignment method of label propagation. On the one hand, the TGNN holistically encodes graph topological information from a multi-modal perspective through a graph convolutional branch (See Section 4.3) and a topological learning branch (See Section 4.4) with a Tensor Transformation Layer (TTL) (See Section 4.2). On the other hand, an MLP classifier is used to produce the predicted labels for source samples and pseudo-labels for target samples, bridging the two components. The source samples are used to calculate the supervised loss while the target samples are used to calculate the consistency regularization of label propagation (See Section 4.5).

---

**Algorithm 1** LP-TGNN

---

1: **Input:** Source graphs $\mathcal{D}^s$; Target graphs $\mathcal{D}^t$.
2: **Output:** GCN parameters $\theta$; CNN parameters $\phi$; MLP parameters $\eta$; TTL parameters $\psi$.
3: **while** not convergence **do**
4:     Randomly sample mini-batches $\mathcal{B}^s$ and $\mathcal{B}^t$ from $\mathcal{D}^s$ and $\mathcal{D}^t$ respectively;
5:     Forward propagate $\mathcal{B}^s$ and $\mathcal{B}^t$ through both branches of TGNN;
6:     Forward propagate the representations of $\mathcal{B}^t$ from each branch, i.e. $\mathcal{Z}^t_{GCN}$ and $\mathcal{Z}^t_{Top}$ through the MLP classifier separately;
7:     Forward propagate the representations of $\mathcal{B}^s$ from both branches, i.e. $\mathcal{Z}^s$ through the MLP classifier;
8:     Calculate the loss by (9);
9:     Update the parameters through backpropagation;
10: **end while**

---

## 4.2. Tensor Transformation Layer (TTL)

The Tensor Transformation Layer: $\sigma\left(\mathcal{L}^{(\ell)}\left(\cdot\right)\right)$ for $\ell = \{1, 2, \cdots, L\}$ preserves the tensor structures of $\boldsymbol{\mathcal{X}}$ of dimension $D = \prod_{m=1}^{M} D_m$. A *deep Tensor Neural Network* is a function in the form of

$$f(\boldsymbol{\mathcal{X}}) = \mathcal{L}^{(L+1)} \circ \sigma \circ \mathcal{L}^{(L)} \circ \sigma \cdots \circ \mathcal{L}^{(2)} \circ \sigma \circ \mathcal{L}^{(1)}(\boldsymbol{\mathcal{X}}) \tag{2}$$

where $\sigma(\cdot)$ is an element-wise activation function. The linear transformation $\mathcal{L}^{(\ell)}(\cdot)$, and input and output tensor of the $\ell$-th layer, i.e. $\mathcal{H}^{(\ell+1)}$ and $\mathcal{H}^{(\ell)}$ are defined by

$$\mathcal{L}^{(\ell)}\left(\mathcal{H}^{(\ell)}\right) := \left\langle \mathcal{W}^{(\ell)}, \mathcal{H}^{(\ell)}\right\rangle + \mathcal{B}^{(\ell)},$$
$$\text{and} \quad \mathcal{H}^{(\ell+1)} := \sigma\left(\mathcal{L}^{(\ell)}\left(\mathcal{H}^{(\ell)}\right)\right) \tag{3}$$

where $\mathcal{H}^{(0)} = \mathcal{X}$ is the input feature tensor, $\langle \cdot, \cdot \rangle$ is the tensor inner product, with a low-rank weight tensor $\mathcal{W}^{(\ell)}$ and a bias tensor $\mathcal{B}^{(\ell)}$. The tensor structure takes effect when we incorporate tensor low-rank structures such as CP, Tucker, and TT.

## 4.3. Graph Convolutional Branch

The convolutional branch learns the representation of $\mathcal{G}$ following the message-passing framework. It utilizes the adjacency matrix $\mathbf{A}$ and feature matrix $\mathbf{X}$ of $\mathcal{G}$ through a sequence of GCN layers [5]. In each layer, each node representation is updated by combining its representation and the aggregated representations of its neighbors from the previous layer. Formally, the representation of $\mathcal{G}$ at the $\ell$-th layer is given by

$$\mathbf{H}_{\mathcal{G}}^{(\ell)} = \sigma\left(\hat{\mathbf{A}}^{\tau} \mathbf{H}_{\mathcal{G}}^{(\ell-1)} \Theta^{(\ell)}\right) \tag{4}$$

where $\hat{\mathbf{A}} = \tilde{\mathbf{D}}^{-\frac{1}{2}} \tilde{\mathbf{A}} \tilde{\mathbf{D}}^{\frac{1}{2}}$, $\tilde{\mathbf{A}} = \mathbf{A} + \mathbf{I}$, and $\tilde{\mathbf{D}}$ is the degree matrix of $\tilde{\mathbf{A}}$. The initial representation is its feature matrix, i.e. $\mathbf{H}_{\mathcal{G}}^{(0)} = \mathbf{X}$, $\mathbf{H}_{\mathcal{G}}^{(\ell)} \in \mathbb{R}^{N \times D^2}$ for $\ell \in \{1, 2, \cdots, L\}$, $\sigma(\cdot)$ is an activation function, and $\Theta^{(\ell)}$ is a trainable weight of the $\ell$-th layer. The $\tau$-th power of the normalized adjacency matrix contains statistics from the $\tau$-th step of a random walk on $\mathcal{G}$, thus enlarging the receptive field of the convolutional operation.

In this branch, different $\tau$-th steps of random walk on $\mathcal{G}$ are combined thanks to TTL, thus boosting the representation power of GCN. Specifically, we first concatenate all representations of the $L$-layer branch to form a tensor denoted by $\mathcal{Z}_{\mathcal{G}}^{GCN} = \left[\mathbf{H}_{\mathcal{G}}^{(1)}, \mathbf{H}_{\mathcal{G}}^{(2)}, \cdots, \mathbf{H}_{\mathcal{G}}^{(L)}\right]$, then feed the tensor into TTL as $\mathcal{H}^{(0)} = \mathcal{Z}_{\mathcal{G}}^{GCN}$, whose $\ell$-th layer is defined in (3). Here $\mathcal{Z}_{\mathcal{G}}^{GCN}$ is reshaped to a dimension of $N \times L \times D \times D$ to facilitate TTL since the output tensors of both branches should have the same number of dimensions (See Section 4.4), as we set the size of $\mathbf{H}_{\mathcal{G}}^{(\ell)}$ as $N \times D^2$ so that $\mathcal{Z}_{\mathcal{G}}^{GCN}$ can be unfolded into four dimensions.

## 4.4. Topological Learning Branch

The topological branch extracts the topological information of a graph $\mathcal{G}$ through Persistent Homology. In detail, to capture the topological information of $\mathcal{G}$, we employ $K$ vertex filtration functions: $f_i : \mathcal{V} \mapsto \mathbb{R}$ for $i = \{1, \cdots, K\}$. Each filtration function $f_i$ gradually reveals one specific topological structure at different levels of connectivity, such as node degree (degree centrality score), node flow information (betweenness centrality score), information spread capability (closeness centrality score), etc. With each filtration function $f_i$, we can construct a set of $Q$ persistence images (PIs) of resolution $P \times P$ from their persistence diagrams.

We further combine the $Q$ persistence images of resolution $P \times P$ from $K$ distinct filtration functions to form a PI tensor denoted by $\mathcal{X}_{\mathcal{G}}$ of size $K \times Q \times P \times P$, which extracts and preserves the multi-modal topological features concealed in $\mathcal{G}$ simultaneously. Note the PIs for a given $\mathcal{G}$ are deterministic, thus to make the branch learnable, we append a convolutional neural network (CNN) layer and a pooling layer to further process the PI tensor. Formally,

$$\mathcal{Z}_{\mathcal{G}}^{Top} = \begin{cases} f_{\text{CNN}}(\mathcal{X}_{\mathcal{G}}) & \text{if } Q = 1 \\ \xi_{\text{POOL}}(f_{\text{CNN}}(\mathcal{X}_{\mathcal{G}})) & \text{if } Q > 1 \end{cases} \tag{5}$$

where $f_{\text{CNN}}$ is a CNN layer and $\xi_{\text{POOL}}$ is a pooling layer such as average pooling or max pooling to ensure a fixed tensor size regardless of the value of $Q$. Similarly, $\mathcal{Z}_{\mathcal{G}}^{Top}$ is then fed into TTL as $\mathcal{H}^{(0)} = \mathcal{Z}_{\mathcal{G}}^{Top}$, whose $\ell$-th layer is defined in (3).

## 4.5. Label Propagation

To achieve domain adaptive graph classification, we further leverage label propagation [26] to regularize both branches. In general, label propagation spreads trustworthy labels obtained from a source task to neighboring data points (possibly in the target domain) through consistency regularization. The underlying intuition is that *neighboring data* should have similar labels. Therefore, its success heavily depends on an appropriate *distance metric* that defines what neighboring data is exactly, which can be particularly challenging to define for structured data, such as graphs. To avoid this obstacle, for a given target sample $\mathcal{G}_j^t$, we consider its neighbor set to be the *source samples* whose *ground-truth labels* are the same as the *pseudo-label* of that target sample, i.e.

$$\Pi_j := \{i | y_i^s = \hat{y}_j^t, \mathcal{G}_i^s \in \mathcal{B}^s\}, \; \mathcal{G}_j^t \in \mathcal{B}^t \tag{6}$$

where $\hat{y}_j^t = \arg\max(q_j^t)$ is its pseudo-label, $q_j^t = \mathrm{MLP}(\boldsymbol{\mathcal{Z}}_j^t)$ is its logits, and $\boldsymbol{\mathcal{Z}}_j^t = \mathrm{Enc}(\mathcal{G}_j^t)$ is its representation from either branch.

Following [26], we implement the consistency regularization from FixMatch [60], i.e.

$$\mathcal{L}_{reg} = \frac{1}{|\mathcal{B}^t|} \sum_{\mathcal{G}_j^t \in \mathcal{B}^t} \mathbb{1}(\max(q_j^t) > \tau) H(\hat{y}_j^t, q_j^s) \tag{7}$$

where $\tau$ is a threshold to ensure that only target samples with sufficiently confident pseudo-labels are considered. $H(\cdot, \cdot)$ is the cross entropy, and $q_j^s = \mathrm{MLP}(\boldsymbol{\mathcal{Z}}_j^s)$. Here $\boldsymbol{\mathcal{Z}}_j^s$ represents the aggregated representations of its neighbor set, e.g. $\boldsymbol{\mathcal{Z}}_j^s = \mathrm{AVG}(\{\boldsymbol{\mathcal{Z}}_i^s\}_{i \in \Pi_j})$, where $\boldsymbol{\mathcal{Z}}_i^s = \mathrm{Enc}(\mathcal{G}_i^s)$. We denote the regularization term from each branch as $\mathcal{L}_{reg}^{GCN}$ and $\mathcal{L}_{reg}^{Top}$ respectively. The two branches are separately regularized as they serve different roles, where the GCN branch plays a major role in graph representation learning, while the other branch provides auxiliary topological information.

For a source sample, we combine its representations from both branches and use it to calculate the supervised loss:

$$\mathcal{L}_{sup} = \frac{1}{|\mathcal{B}^s|} \sum_{\mathcal{G}_i^s \in \mathcal{B}^s} H(y_i^s, q_i^s) \tag{8}$$

Here $q_i^s$ is its logits. Finally, the supervised loss and the two consistency regularization terms are combined to form the overall objective and jointly optimize the whole framework:

$$\mathcal{L} = \mathcal{L}_{sup} + \mathcal{L}_{reg}^{GCN} + \mathcal{L}_{reg}^{Top}. \tag{9}$$

# 5. Experiments

## 5.1. Experiment Setup

**Datasets.** Following [57], we tailor various datasets including MUTAG, Mutagenicity, PROTEINS, DD, BZR, and COX2 from the popular graph classification benchmark TUDataset [1] to the setting of domain adaptation. For convenience, P, D, C, CM, B, BM are respectively short for PROTEINS, DD, COX2, COX2_MD, BZR, BZR_MD. To construct domain discrepancies, we separate Mutagenicity into four same-size subsets (i.e., M0, M1, M2, M3) by edge density quartiles. The details of the datasets are in appendix A.2.

**Baselines.** We validate the performance of the proposed LP-TGNN against various state-of-the-art methods, including one graph kernel method WL Subtree [61], four graph neural network methods including GCN [5], GIN [2], CIN [62], GMT [63], four domain adaptive image classification methods including CDAN [31], ToAlign [64], MetaAlign [65], DUA [66], and two domain adaptive graph classification methods including DEAL [56] and CoCo [57]. Their details are in appendix A.3.

**Implementation details.** We conduct our experiments on one NVIDIA Quadro RTX 8000 GPU card with up to 48GB of memory. The LP-TGNN is trained by the Adam optimizer with a learning rate in {0.01,0.05,0.1}. The hidden dimension and batch size are both set as 32. All activation functions are set

as ReLU. We employ a three-layer GCN in the convolutional branch. For the topological branch, we use four filtration functions, i.e. degree, betweenness, eigenvector, and closeness, set $Q$ in (5) as 1 and the PI dimension as $50 \times 50$. We follow [28] for the choices of tensor low-rank structures. We use the CNN as a 2D convolutional layer followed by a max-pooling layer. The classifier is a two-layer MLP with batch normalization and a dropout rate of 0.5. The threshold for label propagation, i.e. $\tau$ in Eq (7) is set as 0.8. We train the model on all the labeled source samples and all the unlabeled target samples, and evaluate on the ground-truth labels of target samples that are hidden in training by accuracy as in [54]. The code is available at `https://github.com/TaoWen0309/Label-Propagation-GNN`.

## 5.2. Performance Comparison

From Table 1, we observe that 1) LP-TGNN achieves the best performance overall, also LP-GIN reaches the same average accuracy with CoCo and outperforms DEAL; 2) On average, vanilla domain adaptation methods outperform supervised graph learning methods even as they were not developed for graphs, which further demonstrates the low transferability of current graph learning methods. To sum up, our framework tackles domain adaptive graph classification tasks for two key reasons: (i) The high-quality representations produced by GIN and TGNN which are essential for label propagation, and (ii) the efficacy of label propagation in reducing domain discrepancies while maintaining discriminative information, without the strenuous need to define a strict distance between graph samples or a contrived GNN architecture.

Table 1: The classification accuracy (in %) on PROTEINS, COX2, and BZR (source→target).

| METHODS | P→D | D→P | C→CM | CM→C | B→BM | BM→B | Avg. |
|---|---|---|---|---|---|---|---|
| WL SUBTREE | 72.9 | 41.1 | 48.8 | 78.2 | 51.3 | 78.8 | 61.9 |
| GCN | 58.7 | 59.6 | 51.1 | 78.2 | 51.3 | 71.2 | 61.7 |
| GIN | 61.3 | 56.8 | 51.2 | 78.2 | 48.7 | 78.8 | 62.5 |
| CIN | 62.1 | 59.7 | 57.4 | 61.5 | 54.2 | 72.6 | 61.3 |
| GMT | 62.7 | 59.6 | 51.2 | 72.2 | 52.8 | 71.3 | 61.6 |
| CDAN | 59.7 | 64.5 | 59.4 | 78.2 | 57.2 | 78.8 | 66.3 |
| ToAlign | 62.6 | 64.7 | 51.2 | 78.2 | 58.4 | 78.7 | 65.7 |
| MetaAlign | 60.3 | 64.7 | 51.0 | 77.5 | 53.6 | 78.5 | 64.3 |
| DUA | 61.3 | 56.9 | 51.3 | 69.5 | 56.4 | 70.2 | 60.9 |
| DEAL | **76.2** | 63.6 | 62.0 | 78.2 | 58.5 | 78.8 | 69.6 |
| CoCo | 74.6 | 67.0 | 61.1 | **79.0** | 62.7 | 78.8 | 70.5 |
| LP-GIN | 64.0 | **71.2** | 59.1 | 78.2 | **69.3** | **81.0** | 70.5 |
| LP-TGNN | 73.6 | 69.2 | **63.4** | 78.2 | 67.6 | 78.8 | **71.8** |

## 5.3. Ablation Study

To validate the effectiveness of each component in LP-TGNN, we design the ablation experiments on Mutagenicity as shown in Table 2. Specifically, 1) LP-TGNN/TOPO removes the topological learning branch; 2) LP-TGNN/CONV removes the graph convolutional branch; 3) LP-TGNN/SUP removes the supervised loss $\mathcal{L}_{sup}$, and 4) LP-TGNN/LP removes the consistency regularization $\mathcal{L}_{reg}^{GCN}$ and $\mathcal{L}_{reg}^{Top}$. Moreover, we compare the results against the state-of-the-art GIN [2] by applying distribution matching and label propagation to GIN, denoted by DM-GIN and LP-GIN respectively. We also add the results of label propagation with GAT [36] and GraphSAGE [6], i.e. LP-GAT and LP-SAGE.

We have the following observations from Table 2: 1) By comparing DM-GIN and LP-GIN, it is evident that label propagation significantly outperforms distribution matching in this task. Therefore, it is inappropriate to directly apply adversarial-based domain adaptation techniques to graphs. 2) By comparing LP-TGNN/TOPO and LP-TGNN/CONV, we can further verify the different roles of the two branches. In other words, the representation learning of the convolutional branch is crucial, while the topological learning of the other branch provides supplementary information. 3) By comparing LP-TGNN/SUP and LP-TGNN/LP, we can claim that both supervised loss and consistency regularization are indispensable. More interestingly, for either method, the results on

the same target domain are identical for any source domain (except LP-TGNN/SUP on M2→M1), indicating that the framework would lose its transferability without either of the terms. 4) LP-GAT, -GraphSAGE, -GIN have all obtained good performance on several single tasks, while LP-TGNN achieves the best performance overall and ascends especially when M3 is involved, also LP-GIN reaches competitive average accuracy, demonstrating the efficacy of our proposed general framework.

Table 2: The ablation results (in %) on Mutagenicity (source→target).

| Methods | M0→M1 | M1→M0 | M0→M2 | M2→M0 | M0→M3 | M3→M0 | M1→M2 | M2→M1 | M1→M3 | M3→M1 | M2→M3 | M3→M2 | Avg. |
|---|---|---|---|---|---|---|---|---|---|---|---|---|---|
| DM-GIN | 74.6 | 50.3 | 63.3 | 50.3 | 54.8 | 46.3 | 67.0 | 71.8 | 49.8 | 68.6 | 52.8 | 63.4 | 59.4 |
| LP-GIN | 76.0 | 69.3 | 69.6 | 70.0 | 60.2 | **67.9** | 75.9 | **81.6** | 63.7 | 61.3 | 62.3 | 67.7 | 68.8 |
| LP-GAT | 75.4 | 70.2 | 71.2 | **70.8** | 58.8 | 59.3 | 74.5 | 77.4 | 57.8 | 48.3 | 57.6 | 56.7 | 64.8 |
| LP-SAGE | 76.5 | 69.6 | **71.9** | 69.6 | 60.9 | 60.2 | 74.6 | 78.4 | 57.6 | 57.7 | 62.3 | 66.7 | 67.2 |
| LP-TGNN/Topo | 74.7 | 47.7 | 71.3 | 51.0 | 56.5 | 58.6 | 63.5 | 74.8 | 56.4 | 70.0 | 62.2 | 69.2 | 63.0 |
| LP-TGNN/Conv | 32.6 | 61.1 | 37.3 | 55.3 | 53.1 | 55.2 | 65.8 | 72.8 | 53.1 | 32.6 | 53.1 | 62.7 | 52.9 |
| LP-TGNN/Sup | 67.4 | 55.2 | 37.3 | 55.2 | 53.1 | 55.2 | 37.3 | 32.6 | 53.1 | 67.4 | 53.1 | 37.3 | 50.4 |
| LP-TGNN/LP | 67.4 | 44.8 | 62.7 | 44.8 | 46.9 | 44.8 | 62.7 | 67.4 | 46.9 | 67.4 | 46.9 | 62.7 | 55.5 |
| LP-TGNN | **76.6** | **71.9** | 70.8 | 69.7 | **62.9** | 65.2 | 75.7 | 77.1 | **64.9** | **71.5** | **64.2** | **72.8** | **70.3** |

## 5.4. Synthetic Study

The label propagation method was rigorously proved to perform well in computer vision domain adaptation with subpopulation shift [26]. To validate this property in the graph space, we design a synthetic study on MUTAG. Specifically, the negative and positive sample size ratio is approximated to be 1:2 in the source domain and 2:1 in the target domain, constituting a significant subpopulation shift. The performance of the GIN and TGNN methods is listed in Table 3, from which we observe that for the simulated subpopulation shift task: 1) LP-GIN significantly outperforms DM-GIN, achieving competitive performance; 2) Both terms of supervised loss and consistency regularization are important; 3) The topological branch is more pronounced compared with that in Table 2.

## 5.5. Quality of Pseudo-labels

Below we show the quality of pseudo-labels on the Mutagenicity tasks. We report the accuracy for the convolutional branch as it plays a major role in representation learning, and the pseudo-labels are filtered by the confidence threshold first as in (7). We can observe that on average, there exists a positive correlation between pseudo-label and classification accuracy, which further demonstrates the quality of pseudo-labels as the bridge between graph representations and label propagation.

Table 3: The classification accuracy (in %) on the simulated subpopulation shift task.

| Methods | Accuracy |
|---|---|
| DM-GIN | 78.7 |
| LP-GIN | 89.4 |
| LP-TGNN/Topo | 78.7 |
| LP-TGNN/Conv | 80.9 |
| LP-TGNN/Sup | 68.1 |
| LP-TGNN/LP | 89.4 |
| LP-TGNN | **93.6** |

Table 4: The pseudo-label and classification accuracy (in%) on Mutagenicity (source→target).

| Accuracy | M0→M1 | M1→M0 | M0→M2 | M2→M0 | M0→M3 | M3→M0 | M1→M2 | M2→M1 | M1→M3 | M3→M1 | M2→M3 | M3→M2 |
|---|---|---|---|---|---|---|---|---|---|---|---|---|
| **Pseudo-label** | 93.3 | 81.3 | 94.5 | 83.4 | 67.6 | 80.6 | 88.3 | 91.1 | 67.6 | 85.7 | 68.6 | 86.0 |
| **Classification** | 76.6 | 71.9 | 70.8 | 69.7 | 62.9 | 65.2 | 75.7 | 77.1 | 64.9 | 71.5 | 64.2 | 72.8 |

## 6. Conclusion

This work addresses the domain adaptative graph classification problem by proposing a framework named LP-TGNN. Extensive experiment results on various benchmarks validate the efficacy of the proposed LP-TGNN. Besides, the competitive performance of LP-GIN and other baselines showcases the general applicability of our framework. Our work serves as a solid step toward transferable GNNs, as most GNNs so far are trained under full supervision in a single domain. Future work may extend the approach to large-scale datasets and more complex settings, such as multi-label classification and multi-target domain adaptation.

# Acknowledgments

This material is based upon work supported by the U.S. Department of Energy, Office of Science Energy Earthshot Initiative as part of the project "Learning reduced models under extreme data conditions for design and rapid decision-making in complex systems" under Award #DE-SC0024721.

This work was supported in part by the National Science Foundation under Grant DMS-2412577. Any opinions, findings, and conclusions or recommendations expressed in this material are those of the author(s) and do not necessarily reflect the views of the National Science Foundation.

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

# A. Appendix

## A.1. Persistent Images

**Definition 1** (Persistence Image). Let $g : \mathbb{R}^2 \mapsto \mathbb{R}$ be a non-negative weight function for the persistence plane $\mathbb{R}$. The value of each pixel $z \in \mathbb{R}^2$ is defined as $PI_{Dg}(z) = \iint_z \sum_{\mu \in T(Dg)} \frac{g(\mu)}{2\pi\delta_x\delta_y} e^{-\left(\frac{(x-\mu_x)^2}{2\delta_x^2} + \frac{(y-\mu_y)^2}{2\delta_y^2}\right)} dy dx$, where $T(\text{Dg})$ is the transformation of the PD Dg (i.e., for each $(x,y)$, $T(x,y) = (x, y - x)$), $\mu = (\mu_x, \mu_y) \in \mathbb{R}^2$, and $\delta_x$ and $\delta_y$ are the standard deviations of a differentiable probability distribution in the $x$ and $y$ directions respectively.

We can calculate a set of Persistent Diagrams (PDs) for each filtration function $f_i$, i.e., $\text{PH}(\mathcal{G}, f_i) = \overrightarrow{Dg_i} = \{Dg_i^{(1)}, \ldots, Dg_i^{(\mathcal{Q})}\}$, where $\mathcal{Q} \in \mathbb{Z}_0^+$ is the number of graph topological features. Moreover, to encode the above topological information presented in a $Dg$ into the embedding space, we use its vectorized representation, i.e., persistence image (PI) [67]. The PI is a finite-dimensional vector representation obtained through a weighted kernel density function and can be computed in the following two steps. First, we map the PD $Dg$ to an integrable function $\varrho_{Dg} : \mathbb{R}^2 \mapsto \mathbb{R}^2$, which is referred to as a persistence surface. The persistence surface $\varrho_{Dg}$ is constructed by summing weighted Gaussian kernels centered at each point in $Dg$. In the second step, we integrate the persistence surface $\varrho_{Dg}$ over each grid box to obtain the value of the $PI_{Dg}$.

## A.2. Details of Datasets

- MUTAG [29] contains 188 nitroaromatic compounds. The goal is to predict whether a molecule has a mutagenic effect on a given bacterium or not.

- Mutagenicity [68] consists of 4337 molecular structures with their corresponding Ames test data. The goal is to predict whether a molecule is a mutagen or not.
- PROTEINS: PROTEINS [69] and DD [61] contain 1113 and 1178 proteins, where each label indicates whether a protein is an enzyme or not. We denote them in short as P and D respectively.
- COX2: COX2 and COX2_MD [70] are both chemical compounds that consist of 467 and 303 cyclooxygenase-2 inhibitors. We denote them in short as C and CM respectively.
- BZR: BZR and BZR_MD [70] are both chemical compounds that consist of 405 and 306 ligands for the benzodiazepine receptor. We denote them in short as B and BM respectively.

## A.3. Details of Baselines

1. Graph learning methods:
   - WL Subtree [61] presents a family of efficient graph kernels using the Weisfeiler-Lehman test to measure the similarity of graphs.
   - GCN [5] follows the message-passing framework to update node representations iteratively with neighboring nodes.
   - GIN [2] is a state-of-the-art message-passing neural network with increased expressivity by MLPs.
   - CIN [62] extends message-passing Simplicial Networks to regular Cell Complexes and increases the expressivity.
   - GMT [63] is based on multi-head attention and captures the interaction between nodes according to their structural dependencies.

2. Domain alignment methods:
   - CDAN [31] proposes an adversarial learning framework and conditions on the discriminative information from classifier predictions.
   - ToAlign [64] decomposes source domain features into task-related features for alignment and task-irrelevant features to be avoided, based on classification meta-knowledge.
   - MetaAlign [65] treats the optimization of domain alignment and classification tasks as meta-train and meta-test tasks, maximizing the inner product of their gradients during training.
   - DUA [66] proposes an effective and efficient normalization strategy by continuously adapting batch normalization layers.

3. Domain adaptive graph classification methods:
   - DEAL [56] utilizes adversarial learning and adaptive perturbation for domain alignment and distillation for pseudo-labeling.
   - CoCo [57] consists of coupled branches for graph representation learning and contrastive learning between branches and domains.

## A.4. Run-time analysis

To show the computational complexity of LP-TGNN, the running time for the LP-TGNN experiments in Table 2 is given as follows:

Table 5: The running time per epoch (in seconds) of LP-TGNN in Table 2.

| $M_0{\to}M_1$ | $M_1{\to}M_0$ | $M_0{\to}M_2$ | $M_2{\to}M_0$ | $M_0{\to}M_3$ | $M_3{\to}M_0$ | $M_1{\to}M_2$ | $M_2{\to}M_1$ | $M_1{\to}M_3$ | $M_3{\to}M_1$ | $M_2{\to}M_3$ | $M_3{\to}M_2$ |
|---|---|---|---|---|---|---|---|---|---|---|---|
| 5.37 | 4.81 | 5.27 | 4.90 | 4.95 | 4.49 | 4.68 | 4.61 | 4.34 | 4.38 | 4.55 | 4.48 |

