# OpenReview forum: "Bridging Domain Adaptation and Graph Neural Networks: A Tensor-Based Framework for Effective Label Propagation"
_CPAL.cc/2025/Proceedings_Track — CPAL 2025 (Proceedings Track) Poster_

### Official Review · Reviewer_c81E · 2025-01-07

**Rating:** 6
**Confidence:** 3

**Review:**

This paper presents a novel framework, the Label-Propagation Tensor Graph Neural Network (LP-TGNN), which addresses key challenges in domain adaptive graph classification. It effectively integrates tensor-based graph neural network architectures with domain adaptation techniques, primarily through label propagation, to improve the classification accuracy across different graph domains. The framework is tested on a variety of real-world graph classification benchmarks, showing notable performance improvements over existing methods.

Pros:
1.	The paper introduces a unique combination of tensor transformation layers and label propagation to handle domain discrepancies in graph neural networks.
2.	The framework is both theoretically sound and practically effective. The experimental setup is robust, covering multiple comparisons against both traditional and state-of-the-art methods, providing a thorough validation of the model's effectiveness.
Cons:
1.	(Section 5.3) The paper could benefit from a deeper exploration of how the model performs across a more diverse set of domains to evaluate its generalizability. The potential for overfitting to specific graph structures or features tailored by the LP-TGNN architecture needs more detailed analysis, particularly in scenarios where domain variance is subtle.
2.	While the framework shows promising results, the paper lacks a detailed discussion on its computational resources and efficiency, particularly for very large graphs or graph databases with extreme domain variations.

---

### Official Review · Reviewer_ajNu · 2025-01-08
**Using domain adaptation and tensor transformation to improve GNN in graph classification, reasonable model and good results**

**Rating:** 7
**Confidence:** 3

**Review:**

This paper proposes a model called **LP-TGNN**, which involves Domain Adaptation and tensor Transformation to improve GNN performance in graph classification. The proposed GNN  is clear, technically sound, and naturally integrated. The experiments are well-designed, and the results indeed show some improvements.

---

### **Strengths**

1. The paper is well-written and clearly introduces the preliminaries and the authors' contributions

2. Introducing Domain Adaptation for the graph classification problem is natural and reasonable, and is expected to yield substantial performance improvements.

3. The experimental design is well-executed, including  synthetic datasets, real-world datasets, and a comprehensive ablation study.

---

### **Weaknesses**

1. In line 143, to define a domain adaptive graph classification problem,  the authors assume $P_{D^s}(y|X) = P_{D^t}(y|X) $. Does this imply that the problem is only defined when the graphs have the same number of nodes? How would you define it if the graph sizes are different?

2. In the ablation study, the **LP-GIN** model performs very well, with results very close to the proposed LP-TGNN. However, in Figure 2, the authors showed that GIN embeddings are notably different from TTG-NN. Why is there such a discrepancy? Is it because the **label propagation (LP)** component is so crucial that other architectural differences become less significant?

---

Overall, the paper is well-structured, and the contributions are also good.

---

### Official Review · Reviewer_CFfK · 2025-01-10

**Rating:** 7
**Confidence:** 4

**Review:**

**Summary:**

The paper introduces LP-TGNN, a framework to address domain adaptive graph classification by combining tensor-based GNNs with pseudo labelling and label propagation. These three components are all established techniques. The main contribution of the paper is therefore the combination of the three methods and the empirical evidence that the method outperforms competing methods.

**Strengths:**

1. The proposed approach of using label propagation to reduce domain discrepancy (instead of the former adversarial or contrastive learning approaches) is interesting and novel.
2. The method is validated on standard graph benchmarks and demonstrates consistent improvements over several competing methods. There are also ablations showing the individual contribution of each component.
3. The paper is well written, contains helpful visualizations and most sections are easy to follow.
4. The topic of graph domain matching fits well with CPAL as it addresses the important challenge of label scarcity for graph classification.

**Weaknesses:**

1. The section on label propagation is hard to read. Perhaps it would be helpful to explain how it works in general, and then follow up to explain how to use it with the pseudo-labels as a regularization method.
2. More discussion and evidence is needed as to why TP-GNN is the right choice as encoder model. Why is it better at extracting graph representations and preserve graph similarities under its geometric distances than other methods? In the paper there is only a qualitative analysis using t-SNE, which is not sufficient in my opinion. A discussion/comparison with other methods, e.g. the similar two-branch design with a message passing and a graph kernel branch by Luo et al [1], would be interesting.
3. While the experimental section is the most important part of the paper, there are some issues:
    1. A comparison and acknowledgement of the more recent domain adaption method based on adversarial learning DAGRL [1] is missing.
    2. The results of CoCo and DEAL on Mutagenicity are not reported in Table 2. LP-TGNN should get outperformed by them there, but this should not weaken the results too much as it outperforms them on PROTEINS, COX2 and BZR.
    3. The results do not clearly support the claim that TGNN is a better encoder than other GNN methods (see 2.). What does it mean in Table 1 to use, e.g. GIN, here? How does it handle the domain adaptation? In Table 2 you also show results for LP-GIN, which performs quite well. How does this perform for the datasets of Table 1?
    4. Following up on 3.3: What would happen if you chose a stronger GNN encoder model, e.g. sGIN [2]? The results for LP-GIN in Table 2 look already quite promising, what about, e.g. LP-sGIN? Why focus only on LP-TGNN and not on LP-X with LP as a drop-in method to use strong GNN approaches for domain adaptation? (see also final remark)
4. There are some small typos: e.g. references [2] and [59] are identical, in l.201 it should be “In detail” and the “Additionally” in l.55 should be removed, as it currently reads as if there is a third branch in the model.
5. The input arrows and their coloring in Figure 3 are not very clear. I guess you mean both are input into both branches?

**Questions:**

1. What is the intuition for the topological learning branch (apart from including global information)? Is it possible to use it standalone as an alternative to message passing (using the right filtrations)? How do you generally choose the hyperparameters and the filtration functions?
2. Why is it so important to stay in the tensor domain and not just use standard feature maps/channels+CNNs? Why do you also need to combine the output of the two branches in this domain and not just two vectors? Here you refer to a “multi-modal perspective” in the paper, what do you mean by that?
3. I am not an expert for domain adaptation, but how can you refer to the source and target data coming from different data distributions, e.g. on Mutagenicity, if both are samples from the same dataset (Table 2)? Shouldn’t be the datasets more dissimilar for real domain adaption? How different may they be for your/any method to still work?
4. Following up on 3. and more on the philosophical side now: I was thinking about whether your method can be used to learn something really fundamental, e.g. about molecules, in the spirit of a foundation model? What do you think would be needed?

Overall, I think the paper contains a very interesting idea that is worth sharing. However, it would be more convincing if the paper either showed that the label propagation approach improves the domain adaptability with any GNN-based encoder or more clearly motivates and empirically shows why the specific LP-TGNN approach is superior to other methods.

[1] S. Luo, Z. Jiang, Z. Chen and X. Liang, "Domain Adaptive Graph Classification," *ICASSP 2024*

[2] Xinhan Di, Pengqian Yu, Rui Bu, Mingchao Sun, “Mutual Information Maximization in Graph Neural Networks”, *IJCNN 2020*

---

### Official Review · Reviewer_WVhL · 2025-01-15
**a novel framework, LP-TGNN, which aims to tackle the problem of domain-adaptive graph classification**

**Rating:** 6
**Confidence:** 2

**Review:**

This paper presents a novel framework, LP-TGNN, which aims to tackle the problem of domain-adaptive graph classification by combining a tensor-based graph neural network with a label propagation technique originally designed for unstructured data.

Strengths:

(1): The proposed LP-TGNN is novel.

(2) The presentation is clear and easy to follow.

Weakness:

(1) While the tensor-based approach is promising, the paper provides limited discussion about computational overhead for larger-scale datasets.

(2) Although the authors mention low-rank structures like Tucker/CP/TT, they do not thoroughly compare these variants or their impact on performance, leaving open questions on whether any specific factorization structure is best in certain scenarios.

---

### Meta-Review · Area_Chair_SJQN · 2025-02-03

**Recommendation:** Accept (Poster)
**Confidence:** 3

**Metareview:**

All reviewers found the proposed LP-TGNN framework well motivated, novel, and empirically promising for domain-adaptive graph classification. They appreciated easy integration with label propagation and performance gains over baselines. The authors’ strong rebuttal addressed concerns about clarity, missing comparisons, and computational complexity. While some reviewers noted the need for deeper analysis of tensor factorization variants or larger-scale experiments, these are not critical barriers to acceptance. The paper provides a well-founded solution to bridging domain adaptation with GNNs with solid empirical support, and may easily foster further work in this direction.

---

### Decision · Program_Chairs · 2025-02-11

Accept (Poster)